# Correlations between oral Pre-Exposure Prophylaxis (PrEP) initiations and policies that enable the use of PrEP to address HIV globally

**Janki Tailor**[1]☉*, **Jessica Rodrigues**[1]☉, **John Meade**[1], **Kate Segal**[1]☉, **Lilian Benjamin Mwakyosi**[2,3]☉

**1** AVAC, New York, New York, United States of America, **2** COMPASS-Africa, Dar es Salaam, United Republic of Tanzania, **3** DARE Organization, Dar es Salaam, United Republic of Tanzania

☉ These authors contributed equally to this work.
* janki@avac.org

**Data Availability Statement:** Data has been submitted to the UK Data Service data repository. The DOI is 10.5255/UKDA-SN-855973.

## Abstract

Policies facilitating access to HIV prevention services, specifically for pre-exposure prophylaxis (PrEP), can foster enabling environments for service uptake. This analysis aims to establish whether policies enabling broad PrEP eligibility, HIV self-testing, and lowered age of consent to HIV testing and treatment services are correlated with PrEP uptake. Ages of consent vary by country, therefore this analysis focused on how age of consent policies, in general, affect adolescent PrEP uptake. Data was collected from the HIV Policy Lab and AVAC's Global PrEP Tracker, a database of approximately 334 PrEP projects operating across 95 countries, and linear regression and correlation analyses were conducted via STATA to examine relationships amongst national oral PrEP eligibility, HIV self-testing, lowered age of consent, and national cumulative oral PrEP initiations, as of December 2021. Of all 194 countries tracked by the HIV Policy Lab, only about 7% have adopted all three policies (HIV self-testing, lowered age of consent, and PrEP eligibility policies). Less than 50% have adopted have adopted at least one of these policies. Of the 54 countries that have fully adopted PrEP eligibility policies, less than 30% have co-adopted HIV self-testing or lowered age of consent policies. About 30% of these 194 countries have yet to adopt any of these policies, of which about 14% have indicated information is "unavailable" for at least one of the policies. Analyses conducted for the 91 countries tracked by both the HIV Policy Lab and the Global PrEP Tracker revealed a significant and positive relationship between cumulative individuals initiated on oral PrEP and adoption of HIV self-testing policies ($p = 0.01$, $r = 0.26$), lowered age of consent policies ($p = 0.01$, $r = 0.25$), and PrEP eligibility policies ($p = 0.01$, $r = 0.26$). Stronger advocacy efforts towards approving public health policies, such as those outlined in our analysis, that enshrine and enable access to HIV prevention are necessary.

## Introduction

In 2020, there were an estimated 1.5 million new HIV infections globally, bringing the number of people living with HIV to approximately 37.7 million [1]. While new global HIV infections

**Funding:** The authors received no specific funding for this work.

**Competing interests:** The authors have declared that no competing interests exist.

have decreased by 30% in the past decade, they have plateaued over the past five years and there remains an acute and unmet need for HIV prevention services. With HIV pre-exposure prophylaxis (PrEP) able to significantly curb HIV incidence, policies enabling HIV prevention services can aid in reducing such high rates [2].

While there are currently numerous PrEP options in the pipeline, a tenofovir disoproxil fumarate/emtricitabine daily oral pill, or TDF/FTC, has been widely approved as an HIV prevention method [3]. Studies show that oral PrEP can significantly reduce HIV acquisition in individuals across different sexual exposures, genders, and settings [4, 5]. PrEP uptake has been steadily increasing globally, particularly in recent years [6–8]. In 2016, global estimated cumulative PrEP initiations amounted to approximately 102,446. As of December 2021, global estimated cumulative PrEP initiations have grown to approximately 1,984,138 [6]. Some of the greatest increases have occurred in sub-Saharan African countries, first in South Africa, Zambia, and Uganda, with Nigeria, Kenya, Mozambique, Tanzania and Zimbabwe following suit [6]. Brazil and Australia account for the majority of initiations in South America and Oceania, respectively, and Thailand accounts for up to 46% of initiations in Asia [6]. There were over 300,000 global documented initiations in 2020 and 862,000 global initiations in 2021 alone [6, 8].

Policies supporting HIV self-testing can serve to increase oral PrEP uptake. Interventions that aimed to introduce HIV self-testing in low- and middle-income countries (LMICs), such as the STAR initiative [9], found that creating an enabling policy environment increased momentum around HIV self-testing, and, subsequently, demand for oral PrEP. A cohort study in Kenya, where there are full operational guidelines in support of HIV self-testing, also reported that up to 98% of current PrEP users chose HIV self-testing when implemented in-country [10]. Moreover, evidence indicates that HIV self-testing is highly accurate and acceptable [11]. Some benefits include reducing health facility visits, which can save time, foster privacy, and reduce stigma and anxiety about results. Additional benefits include increased perceived security about one's health, economic empowerment, restored intimacy in relationships, and improved protection of one's sexual health [11–13]. Despite barriers associated with HIV self-testing such as the absence of counseling or the fear of partner reactions, randomized clinical trials show that the overwhelming majority of participants choose HIV self-testing over facility-based testing [14–16].

Based on extensive analysis, UNICEF found that it is crucial to lower the age of consent laws for accessing HIV and sexual reproductive health services to ensure that all adolescents who need services can access them, which can be conducive to improving HIV prevention outcomes in this group [17]. Adolescents represent a growing proportion of individuals living with HIV worldwide, with UNICEF reporting that in 2020, approximately 1.75 million adolescents between 10 and 19 were living with HIV globally [18]. In 2020, approximately 400,000 adolescents and young adults between the ages of 10 and 24 were newly infected with HIV [18]. As of July 2021, data suggests that only 25% of adolescent girls and 17% of adolescent boys aged 15–19 had been tested for HIV in the past 12 months in Eastern and Southern Africa, the region most severely impacted by HIV [18]. Testing rates for adolescents and young adults are even lower in Western Africa and South Asia [18]. Lowering age of consent laws for accessing HIV and sexual reproductive health services can help reduce the multiple barriers, including structural barriers, to access and adherence faced by adolescents and young adults compared to adults [19]. Furthermore, legal hurdles for those below the age of legal adulthood, lack of clarity surrounding the need for parental consent and parameters conducive to adolescent consent, and other legal inconsistencies related to HIV health services further impede adolescents' access to PrEP [19]. Age of consent for HIV prevention services varies by country based on definitions established by national governments; policies can apply to

individuals 10 years and older. This analysis only sought to understand the effect of lowered age of consent policies on adolescent access to PrEP.

Despite the effectiveness of oral PrEP and HIV self-testing, the majority of countries have yet to develop and/or implement national policies that authorize broad eligibility of PrEP, HIV self-testing and/or lowered age of consent to initiate HIV testing and treatment services [20]. The role of national policies in improving PrEP uptake is often overlooked and not well-documented, and the relationship between certain policies and HIV prevention outcomes has not been quantified.

## Materials and methods

Understanding the importance of creating a conducive and enabling environment to support PrEP uptake and access to HIV prevention services, the aim of this analysis was to establish whether policies authorizing broad PrEP eligibility, HIV self-testing, and lowered age of consent to HIV testing and treatment services are correlated with PrEP uptake. As national governments set different ages of consent for access to HIV testing and treatment services across countries, this study will focus on how age of consent, in general, affects adolescent uptake of PrEP rather than analyzing age of consent policies for particular countries. Study variables included the adoption status of the above three policies and cumulative PrEP initiations for each of 194 countries. This study used data from the HIV Policy Lab and AVAC's Global PrEP Tracker, a comprehensive database of approximately 334 ongoing PrEP projects and programs operating across 95 countries. Data was collected on policies authorizing eligibility for oral PrEP, self-testing and lowered age of consent to HIV testing and treatment in 194 countries from the HIV Policy Lab, as of December 2021. Each country reported a non-numeric policy status, which was numerically coded for analysis as follows: unavailable was coded as 0, not adopted was coded as 1, planning to submit was coded as 2, under review was coded as 3, partially adopted was coded as 4, and adopted was coded as 5. Data was also collected on the cumulative number of PrEP initiations as of December 2021 for each of the 95 countries in AVAC's Global PrEP Tracker. Linear Regression and Correlations and Covariances analyses were performed using STATA (STATA/SE 17.0) to evaluate relationships amongst national policies that authorize PrEP eligibility, HIV self-testing and lowered age of consent to HIV testing and treatment, and their correlation with oral PrEP initiations. Threshold of significance was p = 0.05.

## Results

Of all 194 countries tracked by the HIV Policy Lab, only 13 (6.7%) have adopted all three policies and 181 (93.3%) have not. Of all countries included, 92 (47.4%) have adopted HIV self-testing policies, 44 (22.7%) have adopted policies authorizing lowered age of consent, and 93 (47.9%) have either fully or partially adopted policies authorizing PrEP eligibility, with only 54 (27.8%) having fully adopted policies authorizing PrEP eligibility. Of these 54, only 44 (22.7% of all countries) have adopted both HIV self-testing policies and policies authorizing broad PrEP eligibility and 15 (7.7% of all countries) have adopted policies authorizing both lowered age of consent for HIV testing and treatment and broad eligibility for PrEP (Table 1, Figs 1–3, respectively). Of all 194 countries, 59 (30.4%) have yet to adopt any policy at all, of which 28 (14.4%) countries have indicated an "unavailable" approval status for at least one of the policies. In this case, countries that had a "partially adopted" status for any of the policies were regarded as having adopted that policy. Linear regression analysis and correlation analysis conducted for the 91 countries tracked by both the HIV Policy Lab and the Global PrEP Tracker (HIV Policy Lab does not track French Guiana and Taiwan) revealed that there is a

**Table 1. Countries policy adoption status, Dec. 2021.**

| Policy | Adopted | Not Adopted | Partially Adopted | Information Unavailable | Total |
|---|---|---|---|---|---|
| Broad PrEP Eligibility | 54 | 73 | 39 | 28 | 194 |
| HIV Self-Testing | 92 | 98 | 0 | 4 | 194 |
| Age of Consent | 44 | 113 | 0 | 37 | 194 |

Of 194 countries, 54 have fully adopted, 39 have partially adopted, 73 have not adopted, and 28 have unavailable information about a broad PrEP Eligibility policy. Of 194 countries, 92 have adopted, 98 have not adopted, and 4 have unavailable information about an HIV Self-Testing policy. Of 194 countries, 44 have adopted, 113 have not adopted, and 37 have unavailable information about an Age of Consent policy.

significant and positive relationship between the adoption of HIV self-testing policies and the estimated number of individuals initiated on oral PrEP in a country ($p = 0.01$, $r = 0.26$). There is also a significant and positive correlation between national policies authorizing lowered age of consent to HIV testing and treatment and oral PrEP initiations ($p = 0.009$, $r = 0.25$), and national policies authorizing PrEP eligibility and oral PrEP initiations ($p = 0.01$, $r = 0.26$)

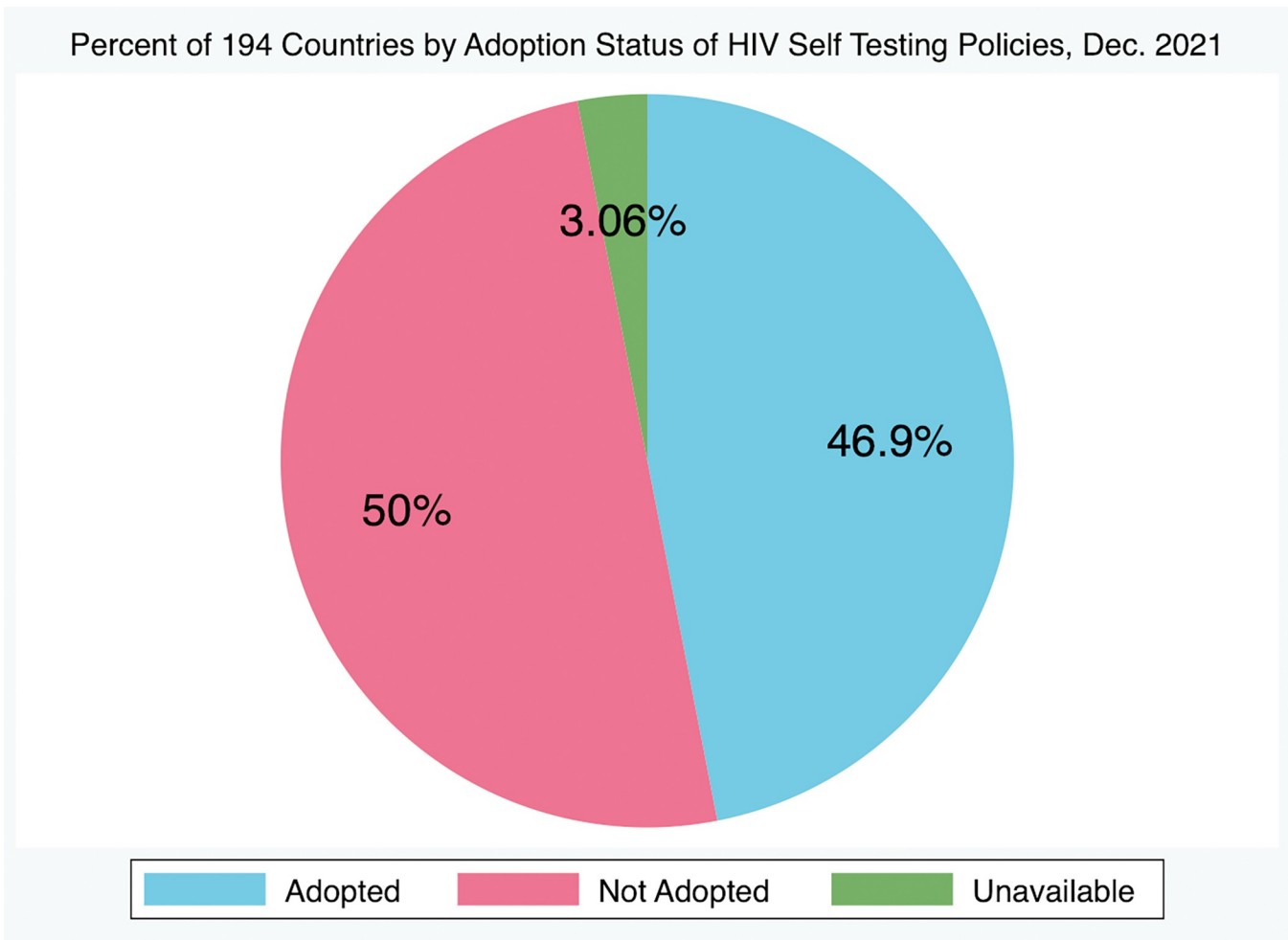

**Fig 1. Percent of 194 countries by adoption status of HIV self testing policies, Dec. 2021.** 46.9% of 194 countries have adopted policies authorizing HIV Self-Testing in-country, 50.0% have not adopted, and information is unavailable for 3.1% of countries, as of December 2021.

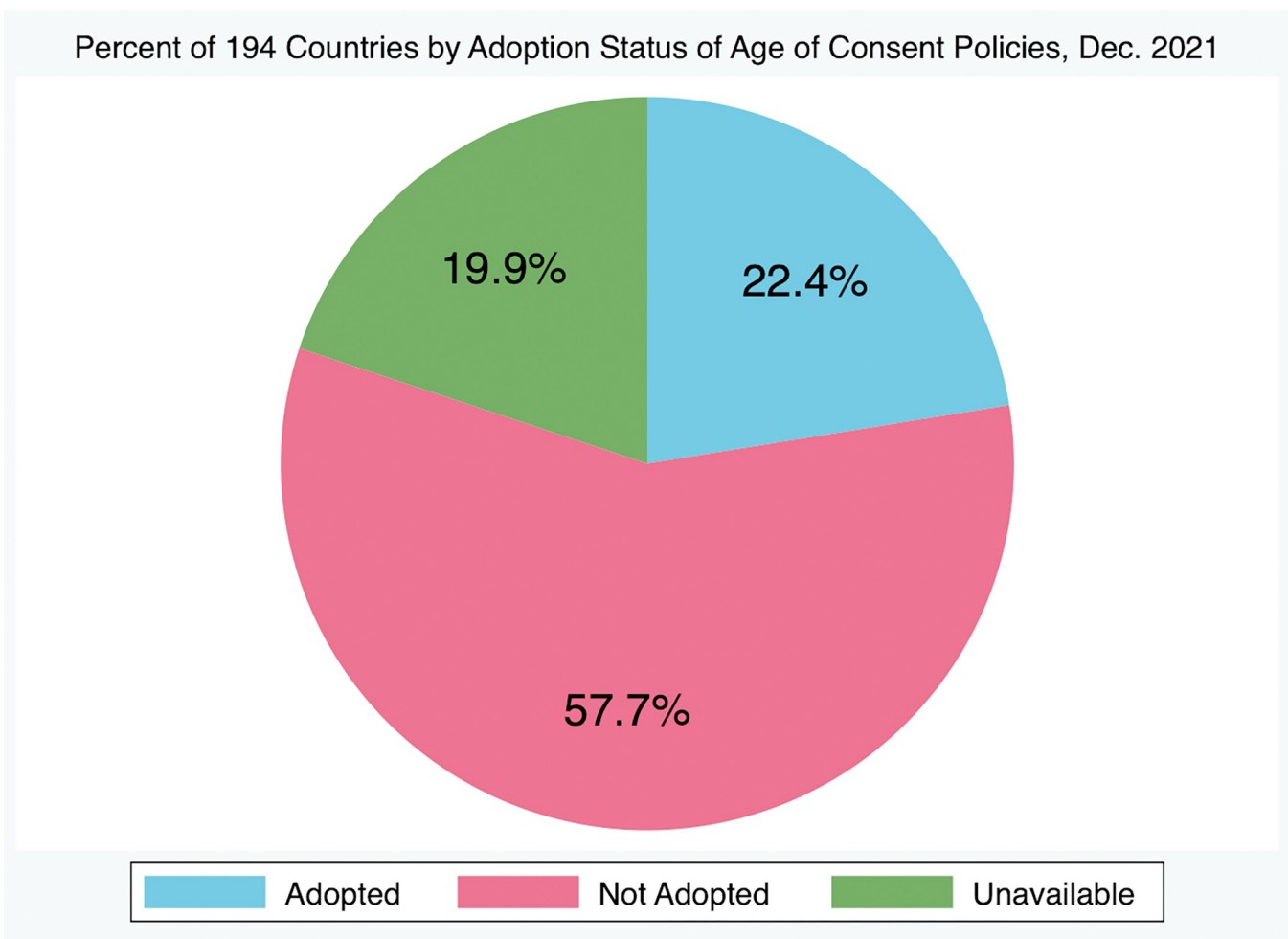

**Fig 2. Percent of 194 countries by adoption status of age of consent policies, Dec. 2021.** 22.4% of 194 countries have adopted policies authorizing lowered age of consent to HIV testing and treatment services in-country, 57.7% have not adopted, and information is unavailable for 19.9% of countries, as of December 2021.

(Table 2). Overall, results showed that policies authorizing broad PrEP eligibility, lowered age of consent, and self-testing all had similar and significant correlations with PrEP uptake.

## Discussion

This study is one of the first to analyze the correlations between policies enabling PrEP use and HIV prevention services. Adoption of policies supporting HIV self-testing, broad eligibility for PrEP, and lowered age of consent for HIV testing and treatment services all strongly correlated with PrEP uptake in-country, with similar significance values. Yet only 13 of 194 countries have adopted all three policies, underscoring a missed opportunity to strengthen access to HIV prevention services.

The significant and positive correlation found between countries authorizing HIV self-testing and cumulative number of PrEP initiations in-country (p = 0.01, r = 0.26) suggests that HIV self-testing is a vital step towards increasing access to PrEP and supports evidence that HIV self-testing serves as a gateway to PrEP uptake [21]. Research shows that this correlation might exist because HIV self-testing mitigates barriers found with facility-based testing,

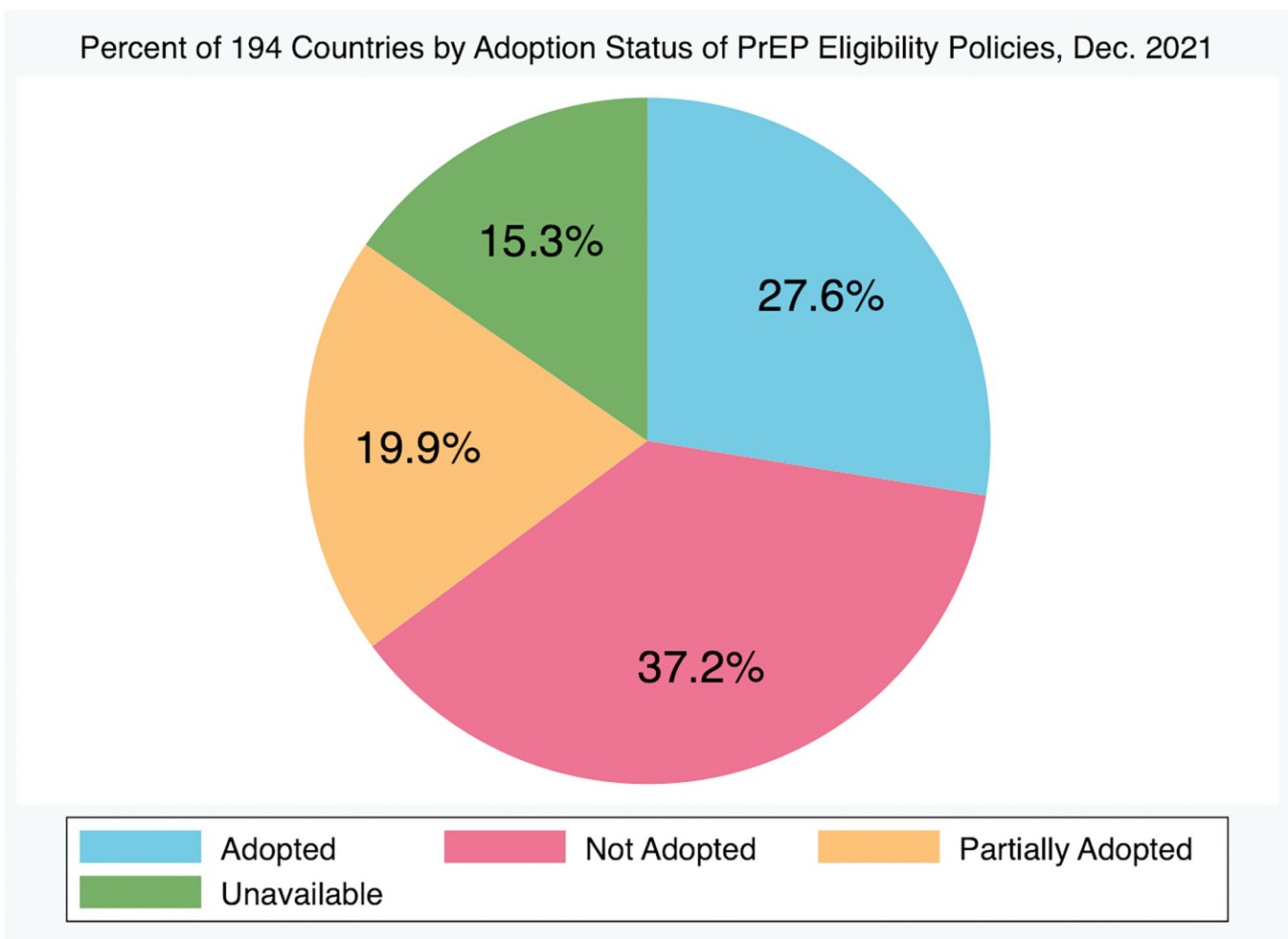

**Fig 3. Percent of 194 countries by adoption status of PrEP eligibility policies, Dec. 2021.** 27.6% of 194 countries have adopted policies authorizing PrEP approval in-country, 19.9% have partially adopted, 37.2% have not adopted, and information is unavailable for 15.3% of countries, as of December 2021.

including reducing time, traveling, costs and stigma involved in determining one's HIV status in a public setting [13, 15, 21]. Given the high acceptability of HIV self-testing, randomized control trials currently underway are seeking to test the effectiveness of using HIV self-testing to support and scale up PrEP delivery in countries such as Kenya [22].

There is also a significant and positive correlation between countries authorizing lowered age of consent to HIV testing and treatment and cumulative number of PrEP initiations (p = 0.01 r = 0.25). This supports findings from studies suggesting that young people are motivated to access PrEP. For instance, studies have found that up to 62% of AGYW in Malawi

**Table 2. Correlation & significance values.**

|  | Broad PrEP Eligibility | | HIV Self-Testing | | Age of Consent | |
|---|---|---|---|---|---|---|
|  | p-value | r value | p-value | r value | p-value | r value |
| **Cumulative PrEP Initiations** | 0.01 | 0.26 | 0.01 | 0.26 | 0.009 | 0.25 |

Adoption of each policy, broad PrEP Eligibility (p = 0.01, r = 0.26), HIV Self-Testing (p = 0.01, r = 0.26), and Age of Consent (p = 0.009, r = 0.25), is positive and significant.

with no perceived risk of HIV acquisition, 69% with low perceived risk, and 78% with high perceived risk are motivated to access PrEP [23]. A study conducted with 825 AGYW in Malawi found that PrEP interest was high among all perceived HIV risk levels, with the odds of showing interest in PrEP being more than doubled in AGYW reporting high perceived risk of HIV acquisition [23]. Another study in Malawi surveying 2,089 adolescents, ages 10–19, and caregivers showed that 82% of adolescents and 87% of caregivers were interested in allowing adolescents access to PrEP [24]. A study in Thailand focused on 200 adolescent men who have sex with men (MSM) and transgender women (TGW) PrEP users, ages 15–19, found that at least 50% of participants were motivated to retain PrEP use six months after intervention due to the presence of youth-friendly services [25]. Such services were the results of Thailand's policies allowing adolescents ages 13–18 access to HIV and STI testing and treatment without parental consent or approval of parental consent exemption by an ethics committee [25].

A significant and positive correlation between countries authorizing lowered age of consent to HIV testing and treatment and cumulative number of PrEP initiations also supports findings from studies that highlight youth perspectives on access to PrEP in the context of parental consent. A study examining the attitudes of sexual and gender minority youths (SGMY) between the ages of 14 and 17 regarding parental or guardian consent to participate in HIV prevention research found that these youth underscored parental permission as a significant barrier to participation [26]. This was especially the case for those fearing physical, psychological or social harm as punishment from parents or guardians opposed to their kids participating in HIV prevention research [26]. Similar results were found in a clinical trial exploring the safety, acceptability and feasibility of PrEP amongst adolescents aged 15–17, where requirements for parental consent were shown to inhibit youth access to and uptake of PrEP due to the unwanted disclosure of sexual activity and sexual orientation [27]. Hence, there is a need and desire for PrEP access amongst youth and policies lowering age of consent may create enabling environments for young people's access to PrEP.

Similarly, and not surprisingly, policies authorizing broad PrEP eligibility also show significant and positive correlations with PrEP uptake (p = 0.01, r = 0.26). Research based on surveillance reports from Belgium suggest that PrEP approval has led to increasing PrEP uptake and has been high ever since [28]. Similarly, research from the U.S. also suggests rapidly increasing PrEP uptake since oral PrEP was approved in 2012 [29]. Population-level influence of PrEP supportive policies is also found in other countries with high levels of oral PrEP uptake and HIV testing and treatment rates. For example, HIV diagnosis rates declined by 35% between 2014 and 2018 in the United Kingdom (UK) and 25% between 2015 and 2019 in New South Wales, Australia despite decreasing levels of condom use [30]. A systematic review that assessed the global trends in the adoption of both WHO PrEP recommendations into national guidelines and the number of PrEP initiations, found that while only 26 countries had adopted the WHO PrEP guidelines by 2016, 120 countries had done so by 2019 [31]. This increase widely corresponded with a global increase in PrEP users across the world [31]. While PrEP users were largely concentrated in the U.S. as of 2015, sizable numbers of PrEP users can be found across the world as of today. Data also indicates that countries that adopted WHO PrEP recommendations earliest were generally also the countries with the greatest cumulative PrEP initiations by 2020, but by the end of 2021, some later adopters had scaled PrEP initiations to similar levels as earlier adopters [6, 8]. However, while PrEP uptake is increasing, the epidemic persists among certain populations such as AGYW in sub-Saharan Africa indicating there may be gaps in access. In global north countries, such as Belgium and the U.S., PrEP uptake has been found to be highest among men and among those between 25 and 50 years of age, with inequity and potential barriers to access having been noted for women and sub-Saharan African migrants [28, 29]. Additionally, while many countries have reported adoption of WHO

PrEP recommendations into national guidelines, this does not necessarily translate into implementation of PrEP services or access to PrEP for all populations in need [31]. These findings enrich our analysis by clarifying that while policies are necessary for fostering enabling environments for HIV PrEP uptake at country level, inequities in PrEP access across populations and settings despite the existence of enabling policies must also be addressed.

While this analysis has found significant correlations between policies enabling PrEP use and HIV prevention services and PrEP uptake, there are limitations. HIV testing is required in order to be eligible for PrEP and therefore this mandate could be a confounding variable [32]. In addition, studies show that HIV testing in general also serves as a gateway to PrEP uptake and we do not know if it is specifically access to HIV self-testing or HIV testing in general that increases PrEP uptake [33].

An additional variable confounding the significant and positive correlation between HIV self-testing policies and PrEP uptake may be self-selection bias. Those with higher levels of risk perception and self-efficacy may be more inclined and motivated to seek HIV testing and access PrEP [34, 35].

Another limitation is that age of consent policies can often be quite broad and cover a range of health interventions without describing specific regulations related to oral PrEP access [36]. In other words, these policies may not explicitly prohibit PrEP access for adolescents but may implicitly do so by limiting providers' ability to provide PrEP due to lack of clarity on whether PrEP provision for minors is legal. The interpretation of laws authorizing age of consent varies widely across countries and may impact PrEP uptake differently.

Additional limitations of this analysis include that this analysis uses secondary sources and that the data sources used for this analysis were missing data points for some countries.

Despite these limitations, our findings underscore that enabling policy environments can foster greater PrEP uptake. Governments and advocates can leverage this evidence to promote adoption of enabling policies for HIV prevention services, including policies covered in this analysis, as a tool to address the HIV epidemic.

## Conclusion

While the adoption of PrEP recommendations into national guidelines exists and is a vital step towards increasing PrEP availability in-country, much more needs to be done to implement effective PrEP services. Despite the fact that policies permitting HIV self-testing, PrEP eligibility, and lowered age of consent for HIV testing and treatment are significantly correlated with PrEP uptake, many countries still do not have some or all of these policies in place. Results from this analysis imply that enabling policy environments have direct links to PrEP uptake and that policies facilitating access to HIV prevention services will be vital to effectively contain the HIV epidemic. With several additional PrEP options in the pipeline, equitable access will depend on a policy landscape that includes diverse PrEP choices and needs and breaks down barriers to engage in HIV prevention services. Civil society plays a vital role in identifying the needs of the most vulnerable and protecting their rights. Therefore, when considering future policy changes or adoption of new PrEP options, governments and civil society must continue to actively collaborate and strengthen joint efforts to effectively control the HIV epidemic.

## Acknowledgments

We thank Mitchell Warren of AVAC for contributing critical thoughts and edits to this research.

## Author Contributions

**Conceptualization:** Janki Tailor, Jessica Rodrigues, John Meade.

**Data curation:** Janki Tailor.

**Formal analysis:** Janki Tailor.

**Methodology:** Janki Tailor, Jessica Rodrigues.

**Software:** Janki Tailor.

**Supervision:** Jessica Rodrigues.

**Visualization:** Janki Tailor.

**Writing – original draft:** Janki Tailor.

**Writing – review & editing:** Janki Tailor, Jessica Rodrigues, John Meade, Kate Segal, Lilian Benjamin Mwakyosi.

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
