## [Decision Letter · Decision Letter 0]

17 May 2022

PGPH-D-22-00733

Policies influence practice: national policies and their correlations with oral prep initiation rates

Dear Janki Tailor,

Thank you for submitting your manuscript to PLOS Global Public Health. After careful consideration, we feel that it has merit but does not fully meet PLOS Global Public Health’s publication criteria as it currently stands. Therefore, we invite you to submit a revised version of the manuscript that addresses the points raised during the review process.

We look forward to receiving your revised manuscript.

Kind regards,

Collins Otieno Asweto, PhD

Academic Editor

Journal Requirements:

1. Please provide an Author Summary. This should appear in your manuscript between the Abstract (if applicable) and the Introduction, and should be 150–200 words long. The aim should be to make your findings accessible to a wide audience that includes both scientists and non-scientists. Sample summaries can be found on our website under Submission Guidelines:

https://journals.plos.org/globalpublichealth/s/submission-guidelines#loc-parts-of-a-submission

**Comments to the Author**

1. Does this manuscript meet PLOS Global Public Health’s publication criteria? Is the manuscript technically sound, and do the data support the conclusions? The manuscript must describe methodologically and ethically rigorous research with conclusions that are appropriately drawn based on the data presented.

Reviewer #1: Yes

Reviewer #2: Partly

2. Has the statistical analysis been performed appropriately and rigorously?

Reviewer #1: No

Reviewer #2: Yes

3. Have the authors made all data underlying the findings in their manuscript fully available (please refer to the Data Availability Statement at the start of the manuscript PDF file)?

Reviewer #1: No

Reviewer #2: No

4. Is the manuscript presented in an intelligible fashion and written in standard English?

Reviewer #1: Yes

Reviewer #2: Yes

5. Review Comments to the Author

Reviewer #1: The authors should clarify and show in tables statistical analysis of correlations between estimated number of cumulative individuals intitiated on oral PrEP and adoption of HIV self-teting policies, adoption of lowered age of consent policies, and adoption of PrEP eligibility policies.

Please remove figures (graphes) you presented in results and instead of this put those tables with realtionships and statistical siginficance values of tests.

Also, data in graphes explaine in text of results.

Reviewer #2: After careful analysis of the study entitled: “Policies influence practice: national policies and their correlations with oral prep. initiation rates”, the authors expose a relevant and current topic.

As I followed the reading of the entire manuscript, I left a suggestion for a more appropriate title change.

Initially in the Abstract sections, as a reader, I felt the lack of some basic elements that should be included in these; I leave suggestions for changes in the comments of the manuscript file.

The Introduction is good, however I leave some suggestions for improvement, such as summarizing it, it is very extensive, and adjustments regarding the argumentation and objective of the study, as well as reviewing the scientific writing of excerpts that can be better used in other sections of the manuscript.

In general, the study needs to be revised in terms of its scientific writing, especially in the Summary and Methodology sections.

The study methodology is adequate, but needs to be more detailed; some details that the authors are likely to know do not appear to readers who are thus harmed as to use in future studies for a scientific comparison.

The Results are very comprehensive and only graphics do not convey the due importance of the findings. I leave as a suggestion the use of tables also to enrich the work scientifically, as well as its greater detail to be better used by the reader.

The Discussion developed by the authors has potential. A better development of this section is necessary, with arguments ending some paragraphs. In the end, the Discussion was not carried out; there is a lot of information and results that should be better finalized, as well as the potential of the study explained. A good final justification for the importance of this study being carried out must necessarily close this section, since the authors presented so many possible biases and limitations.

The Conclusion of the study is presented as a part that can be included in the Discussion of the same, in my opinion.

Finally, in the end, the article should show readers what the purpose was, despite what policies allow for HIV self-testing around the world, and the results showing that favorable policy environments have direct links with the adoption of PrEP, as well as the importance of policies that facilitate access to HIV prevention services to effectively contain an epidemic. Below are comments and suggestions for revision and corrections.

I congratulate the authors for choosing the topic and for the work carried out. I recommend publishing after making the suggested adjustments and corrections.

I would like to have access to the article after pre-publication authors' adjustment.

Thank you for the opportunity to learn and exchange knowledge.

Cordially, good work and success to all!

Comments/Suggestions

Title:

Suggestion: change the more appropriate title: "Correlations between Initiation and Oral Preparation Policies that enable the use of pre-exposure prophylaxis to HIV worldwide"

Lines; 33-34

If: "Only 13 (6.7%) have adopted all three policies..." How: "many countries have yet to adopt any or all three policies"????

Controversial!

Suggestion: review here and in the Results section!

Line: 41

What about Public Health Policies???

lines: 20 - 41

Very extensive introduction. Suggestion: summarize. Describe: which continents evaluated, in the world. What are the study variables? How long the research took and how it was carried out (tools used). In conclusion: include that public health policies, in this sense, should be reanalyzed and/or implemented.

Line: 42

Incluir palavras chaves

include keywords

Consult the Decs and adjust the words...

HIV Prevention; oral pre-exposure prophylaxis; HIV PrEP; PrEP uptake; policy

environment; HIV self-testing; Age of consent; PrEP eligibility; PrEP regulatory

approval; PrEP guidelines

Line: 45

Delete word.

Lines: 80-96

The focus of the study was which age group?

Suggestion: define and insert which age group is defined in the Abstract, here in the Introduction focus on the chosen age(s) and in the Methodology make it clear as well.

Lines: 98 - 110

This whole part is characterized more as Discussion of the manuscript. Suggestion: remove and include in the Discussion section. And here, if you want to keep the information, summarize it well and objectively, putting only the source.

Lines: 117-119

Suggestion: insert this objective in the Summary.

Lines: 119-120

That's methodology. Suggestion: remove and insert in the Methodology section.

Line: 123

Attention to Scientific Writing when writing articles. Suggestion: review Scientific Writing throughout the manuscript!

Lines: 123-25

Suggestion: insert information in the Summary

Line: 125

Attention to Scientific Writing when writing articles. Suggestion: review Scientific Writing throughout the manuscript!

Lines: 125-31

This methodology was based on studies already carried out. Suggestion: if positive, cite source.

Line: 131

Attention to Scientific Writing when writing articles. Suggestion: review Scientific Writing throughout the manuscript!

Line: 137

Suggestion: Graphs illustrate briefly and objectively, however, considering the number and scope of information, there is a need to build at least one very informative table that includes all the information found.

Mainly to show the reader, in detail, how it was carried out and what findings the study reached through linear regression and correlation analyzes.

Lines: 154-58-62

How many countries in total? Year??

Lines: 169-71

Is this the main result of the study? If there are more main results, they should be highlighted here in the first paragraph of the Discussion (Scientific Writing).

Lines: 200-24

With this, it is concluded, in this paragraph, what from these findings compared to those of the study in question? Suggestion: contest...

Line: 247

In contrast, what would be the potential of these findings? Suggestion: include as potentialities of this study.

Line: 248

Suggestion: End the Discussion with this conclusion:

"250 While the adoption of PrEP recommendations in national guidelines is a vital step towards increasing the availability of PrEP in the country, much more needs to be done to implement PrEP services. With several additional PrEP options in the pipeline, equitable access will depend a policy landscape that includes diverse PrEP choices and needs and breaks down barriers to engage in HIV prevention services. Civil society plays a vital role in identifying the needs of the most vulnerable and protecting their rights. Therefore, when considering future policy changes or adoption of new PrEP options, governments and civil society must continue to collaborate and strengthen joint efforts to effectively control the HIV epidemic."

Line: 249

Suggestion: carry out an objective conclusion. Suggestion: carry out an objective conclusion

Suggestion: Conclusion

Despite the fact that policies that allow for HIV self-testing, eligibility for PrEP, and lowering the age of consent for HIV testing and treatment are significantly correlated with PrEP, many countries still do not have some or all of these policies in place. results of this imply that enabling policy environments have direct links to PrEP adoption and that policies that facilitate access to HIV prevention services will be vital to effectively contain HIV epidemic.

Line: 250

Suggestion: incluide the word "exist"

Line: 268

Suggestion: review references and include "DOI", if any.

6. PLOS authors have the option to publish the peer review history of their article (what does this mean?). If published, this will include your full peer review and any attached files.

**Do you want your identity to be public for this peer review?** For information about this choice, including consent withdrawal, please see our Privacy Policy.

Reviewer #1: No

Reviewer #2: No

---

## [Decision Letter · Decision Letter 1]

1 Nov 2022

Correlations Between Oral Pre-Exposure Prophylaxis (PrEP) Initiations and Policies that Enable the Use of PrEP to Address HIV Globally

PGPH-D-22-00733R1

Dear Janki,

We are pleased to inform you that your manuscript 'Correlations Between Oral Pre-Exposure Prophylaxis (PrEP) Initiations and Policies that Enable the Use of PrEP to Address HIV Globally' has been provisionally accepted for publication in PLOS Global Public Health.

Best regards,

Collins Otieno Asweto, PhD

Academic Editor

Reviewer Comments (if any, and for reference):

Reviewer's Responses to Questions

**Comments to the Author**

1. If the authors have adequately addressed your comments raised in a previous round of review and you feel that this manuscript is now acceptable for publication, you may indicate that here to bypass the “Comments to the Author” section, enter your conflict of interest statement in the “Confidential to Editor” section, and submit your "Accept" recommendation.

Reviewer #1: All comments have been addressed

Reviewer #2: All comments have been addressed

2. Does this manuscript meet PLOS Global Public Health’s publication criteria? Is the manuscript technically sound, and do the data support the conclusions? The manuscript must describe methodologically and ethically rigorous research with conclusions that are appropriately drawn based on the data presented.

Reviewer #1: Yes

Reviewer #2: Yes

3. Has the statistical analysis been performed appropriately and rigorously?

Reviewer #1: Yes

Reviewer #2: Yes

4. Have the authors made all data underlying the findings in their manuscript fully available (please refer to the Data Availability Statement at the start of the manuscript PDF file)?

Reviewer #1: Yes

Reviewer #2: Yes

5. Is the manuscript presented in an intelligible fashion and written in standard English?

Reviewer #1: Yes

Reviewer #2: Yes

6. Review Comments to the Author

Reviewer #1: (No Response)

Reviewer #2: I congratulate the authors for choosing the topic and for the work carried out. After making the suggested adjustments, with considerable improvement in the quality of the manuscript, I recommend publication.

Thank you for another opportunity to learn and exchange knowledge. I remain available for further revisions!

Cordially, good work and success to all!

7. PLOS authors have the option to publish the peer review history of their article (what does this mean?). If published, this will include your full peer review and any attached files.

**Do you want your identity to be public for this peer review?** For information about this choice, including consent withdrawal, please see our Privacy Policy.

Reviewer #1: No

Reviewer #2: **Yes: **Carla Fabiana Tenani
